# Effects of a Novel Lin Seed Polysaccharide on Beef Sausage Properties

**DOI:** 10.3390/polym15041014

**Published:** 2023-02-17

**Authors:** Aicha Chouikhi, Naourez Ktari, Sirine Ben Slima, Imen Trabelsi, Farida Bendali, Riadh Ben Salah

**Affiliations:** 1Laboratory of Biotechnology Microbial Enzymatic and Biomolecules (LBMEB), Center of Biotechnology of Sfax, Sfax 3018, Tunisia; 2Laboratory of Enzyme Engineering and Microbiology, National School of Engineering of Sfax (ENIS), Sfax 1173-3038, Tunisia; 3Department of Life Sciences, Faculty of Science of Gabes, Gabès 6029, Tunisia; 4Laboratoire de Microbiologie Appliquée, Faculté des Sciences de la Nature et de la Vie, Université de Bejaia, Bejaia 06000, Algeria

**Keywords:** functional properties, antioxidant activities, quality of sausages

## Abstract

Functional ingredients are substances that offer health benefits beyond their nutritional value. A novel heteropolysaccharide, named *Linum* water soluble polysaccharide (LWSP) was purified from *Linum usitatissimum* L. seeds powder and identified, via TLC and NMR, as a polymer composite of α1-2-L-arabinose, β1-2-D-xylose, β1-2-D-mannose and α1-2-D-glucose. The effect of incorporating LWSP on the quality of beef sausages, stuffed into collagen casings after 15 days of storage at 4 °C, was evaluated for texture profile analysis, color, sensory analysis and oxidation attributes. The new sausages formulated with LWSP recorded good textural attributes via reduction of cohesiveness, hardness and chewiness and improved the sensory features, especially texture, color and general acceptability. In addition, substituting ascorbic acid, a synthetic antioxidant, via the biological ingredient LWSP, retarded lipid oxidation and improved the oxymyoglobin rate until 15 days of storage. LWSP was proved to be a good natural substituent to synthetic antioxidants that definitely improves the oxidation stability and quality of sausages.

## 1. Introduction

Food researchers and manufacturers continuously seek healthier food formulations using functional ingredients as natural preservatives and antioxidant agents [1]. In the meat industry, new formulations have allowed functional components to be incorporated and potentially harmful ingredients to be removed or minimized [2,3]. Meat products are delicate due to the vulnerability of their chemical and biological characteristics. In fact, myoglobin and lipid oxidations lead to sensorial degradation and an unpleasant brown coloration of meat products. Denaturation of the globin exposes the heme group and increases the susceptibility of heme to oxidation. The heme group contains an iron atom that can exist in a reduced (ferrous/Fe^2+^) or oxidized (ferric/Fe^3+^) form [4].

Synthetic antioxidants such as BHA (butyl hydroxy anisole) and BHT (butyl hydroxy toluene) are commonly added to delay meat lipid and myoglobin oxidation. However, they are inadvisable due to their carcinogenic potential [1]. Meanwhile, natural antioxidants not only offer extra health benefits, but also extend meat products’shelf-life and improve their nutritional quality [5,6]. Among these, polysaccharides have recently been attributed special attention thanks to their various biological activities. They have several positive health effects and could be used to protect the human body against free radicals and delay the risk of various chronic diseases including cancer, heart disease, arthritis and aging [7]. Researchers also reported that polysaccharides’ functional properties are interesting for their impact on meat product quality. Indeed, they are generally used in meat products as emulsifiers [6]. They also have fat and water holding, texture improvement shelf-life extension capacities [8]. A sulfated polysaccharide from red macro alga *Falkenbergiarufolanosa* (FRP) incorporated in sausages induced depletion in lipid and myoglobin oxidation and in free fatty acids contents as well as, decreasing the microbial counts [9].

*Linum usitatissimum* L. belongs to the Linaceae family, commonly known as flax [10]. It is cultivated in moderate climate. Linseed is considered as a functional ingredient thanks to its nutritional potential and abundance in biologically active components [11,12,13]. Indeed, it can reduce the content of serum cholesterol and the tumor growth rate and the prostate, breast and colon cancer incidences [14,15,16]. In fact, it has been reported to possess several antioxidants and these are potential ingredients in pharmaceutical, cosmetic and foods compositions.

Linseed is full of antioxidants such as tocopherols, betacarotene, cysteine and methionine, which result in a decrease in blood pressure, heart disease, hepatic and neurological disorders and increased insulin sensitivity. Flaxseed is commonly used for its antidiabetic and anticancer activities and also it is beneficial for cardiovascular, gastrointestinal, hepatic, urological and reproductive disorders, and because of these beneficial effects, it is recognized as a medical plant. Furthermore, it is important in the food chain throughout the world because of its physiological advantages in preventing or treatment of diseases as a functional food. Flaxseed has been regarded as a generally recognized as safe (GRAS) source of vitamins, minerals, proteins and peptides (including bioactive cyclic peptides), lipids (including omega-3 and omega-6 polyunsaturated fatty acids), carbohydrates, lignans and dietary fiber. Due to its beneficial features for health, it has been applied in specific diets [17]. Flaxseed polysaccharides are a heteropolysaccharide composed of neutral and acidic monosaccharides. The differences in monosaccharide composition of flaxseed have been attributed to differences in cultivars, the extraction process and the temperature of extraction [18]. Polysaccharides flaxseeds present important functional properties and biological activities. These properties presented its application in many fields. As the main bioactive component, flaxseed polysaccharide could improve lipid metabolism including the improvement of lipolysis and the suppression of lipogenesis and prevent metabolic syndrome by modulating the gut microbiota [19]. In addition, these biopolymers can increase the viscosity of salad dressings, juices and bakery products. Its high water and oil binding capacities can be applied to improve the texture of meat and bakery products, whereas emulsifying properties of both hydrocolloids improve the firmness and elasticity of these products, as well as the stability of oil and water emulsions [20]. 

In this study, a novel polysaccharide was extracted from Lin seeds, named *Linum* water soluble polysaccharide (LWSP) and characterized via TLC, NMR. In addition, physico-chemical analysis (carbohydrate, protein, fat, moisture and ash contents) and functional properties (water and oil holding capacities) were studied. Its application in beef sausages may improve the quality characteristics of the final product. The aim of this work is to study the effect of the incorporation of LWSP in beef sausages on lipid and myoglobin oxidation, as well as physico-chemical, color, texture and sensory characteristics of the formulated sausages after 15 days of storage at 4 °C.

## 2. Material and Methods

### 2.1. Material and Reagents

*Linum usitatissimum* (L.) seeds were obtained from a local market of Sfax city in Tunisia. The material was ground with a Moulinex blender LM 241 and then sieved with a 0.2 mm mesh size. Round beef meat and ingredients were obtained from CHAHIA Company (Sfax, Tunisia). Round comes from the inside of the leg and is therefore sometimes called inside round. In its retail form, it usually consists of two muscles, the semimembranosus and the adductor.

Meat was minced (through a 6 mm grinding plate) using a mincer machine (FP 3010 food processor, BRAUA, Frankfurt, Germany). It was stored at −20 °C (WHIRLPOOL, serial number WTE2921) until use in sausage formulation.

### 2.2. Extraction of LWSP

LWSP extraction was performed following Ben Slima et al. [5]. At room temperature, pre-extraction of *Linum usitatissimum* (L.) seed powder was made with 95% ethanol to discard pigments. The residue extraction was performed twice with water (20 volumes) with heating at 90°C for 4 h. After the vacuum evaporation of filtrates, precipitation with 95% (*v*/*v*) ethanol was applied to the concentrated liquid over 24 h at 4 °C followed via centrifugation for 15 min at 2268× *g* using a centrifuge (Hettich Zentrifugen, ROTINA 380R, Germany). Afterward, LWSP was obtained at 50 °C.

### 2.3. Thin-Layer Chromatography (TLC)

LWSP was hydrolyzed in trifluoroacetic acid (TFA) (4 M) over 8 h at 100 °C. Then, the obtained hydrolysis LWSP was analyzed via TLC. A mixture with a ratio of 6:7:1 (*v*/*v*) of chloroform/acetic acid/water was used as a developing solvent for released sugars. Spot was visualized via the spraying of 5% (*v*/*v*) H_2_SO_4_ in ethanol incubation and oven drying was performed at 105 °C for 10 min. Glucose, galactose, mannose, xylose, tagatose, rhamnose, fructose, and arabinose served as standard.

### 2.4. NMR Spectroscopy

NMR experiments of LWSP were recorded on a Bruker 400 spectrometer (Bruker Biospin AG, Fallanden, Switzerland) at 25 °C. LWSP was dried and exchanged with deuterium by lyophilizing with D_2_O. The deuterium-exchanged polysaccharide (20 mg) was put in a 5 mm NMR tube and dissolved in 1 mL 99.9% D_2_O. ^13^C NMR and ^1^H NMR spectra were recorded at 75.5 and 400 MHz, respectively. Data analysis was carried out using MestRe Nova 5.3.0 (Mestrelab Research S.L.) software.

### 2.5. Physico-Chemical Analysis of LWSP

A Color Flex spectrocolorimeter (Hunter Associates Laboratory Inc., Reston, VA, USA) served to determine L*, a* and b* values of LWSP color. The moisture and ash contents were evaluated according to the AOAC standard methods 930.15 and 942.05, respectively. Crude protein and fat contents and total carbohydrates were determined according to Ben Slima et al. [5].

### 2.6. Functional Properties of LWSP

#### 2.6.1. Water Holding Capacity

The water holding capacity (WHC) was recorded using the method of Trigui et al. [21]. One gram of LWSP was dispersed in distilled water (25 mL). The suspensions were held at different temperatures (25, 50 or 75 °C) for 1 h. The supernatant was then carefully decanted and excess water was discarded via draining at 50 °C for 25 min. The WHC was calculated according the following formula:

The WHC was expressed as g of absorbed water per g of sample.
$$\mathrm{WHC}=\frac{\mathrm{Weight}\;\mathrm{of}\;\mathrm{the}\;\mathrm{tube}\;\mathrm{content}\;\mathrm{after}\;\mathrm{draining}}{\mathrm{Weight}\;\mathrm{of}\;\mathrm{the}\;\mathrm{dried}\;\mathrm{LWSP}}$$

#### 2.6.2. Oil Holding Capacity (OHC)

Oil holding capacity (OHC) was determined following Trigui et al. [21]. The samples (0.5 g) were poured in volumes of 10 mL of soybean oil. Then, the suspensions were stirred, kept at different temperatures (25, 50 or 75 °C) for 1 h and centrifuged at 7168× *g* for 20 min. The obtained supernatant was removed and the centrifuge tube was drained. OHC was calculated as the weight of the tube contents after draining divided by the weight of the dried LWSP and expressed as g of absorbed oil per g of sample.

### 2.7. Beef Sausage Product Preparation

To prepare sausage, meat was thawed (18 h at 4 ± 2 °C) and prepared as described by Ben Slima et al. [2]. Meat was manually homogenized, and half of the additives and water were then added while mixing. The second half of the ingredients was then added and the mixture was homogenized. Immediately, sausages were manually stuffed into 27 mm diameter reconstituted collagen casings with a length of 15 cm obtained from Qingdao Ifine Casing Co., Ltd. (Qingdao, China). After that, sausages underwent heat processing in a waterbath Haake, Kalsruhe (Germany) at 90 °C until the internal temperature reached 74 °C. Ultimately, sausages were cooled and stored aerobically in refrigerators at 4 °C (WHIRLPOOL, serial number WTE2921) in the absence of light until analysis.

We used 195 g of meat to prepare 300 g of sausage per formulation. We made four formulations (300 g for each formulation) so the meat used was 780 g. The sausage formulation included meat (65%) and the following ingredients: modified starch (8.48%), NaCl (1.29%), sodium tripolyphosphate (0.31%), carrageenan (0.69%), colorant (0.01%), NaNO_2_ (0.01%), cold water (23%) and ascorbic acid or LWSP. Four treatments of sausage were performed: T1: control sausage containing neither LWSP nor ascorbic acid (water: 23%+0.125%); T2: standard formulation consisting of 0.125% ascorbic acid; T3: sausage reformulated with LWSP at 0.125%; and T4: sausage reformulated with LWSP at 0.062% and ascorbic acid at 0.062%. Beef sausages were formulated in triplicate.

### 2.8. Analysis of Sausage Samples

#### 2.8.1. Textural Profile Analysis

Textural profile analysis (TPA) of sausage samples was done according to Ayadi et al. [22] with a texture analyzer (TA-XT2i, Stable Micro Systems Ltd., Surrey, UK). Cylinders of 2 cm in diameter and 2 cm in length, at a temperature of 25 °C, were prepared from each sample and compressed twice to 50% of their original height between flat plates and a cylinder. The following TPAs were determined: hardness (N), springiness (mm), cohesiveness and chewiness (Nmm). To evaluate stability, TPAs were measured after 1 and 15 days of storage at 4 °C.

#### 2.8.2. Color Measurement

Color was measured using a color flex spectrocolorimeter (Hunter Associates Laboratory Inc., Reston, VA, USA, Illuminant D65, 2.54 cm diameter aperture, 10° standard observer) after 15 days of storage to determine CIELAB values: L*, a*, and b*. The L* value referred to color lightness (0, 100), the a* value the span of green-red color (−100, +100) and the b* value the extent of blue-yellow color (−100, +100). The treatments were realized in triplicate samples and the determination of color was carried out in triplicate. Colors were measured at the first day and after 1 and 15 days of storage on cut sausage slices. 

The total color change (ΔE) was then calculated for each sample, using the equation below: ΔE = [(ΔL*)^2^ + (Δa*)^2^ + (Δb*)^2^] ^0.5^
where ΔL *, Δa * and Δb * are the derivatives of corresponding parameters, respectively.

#### 2.8.3. Thiobarbituric Acid Reactive Substances (TBARS) Analysis

Lipid oxidation in different samples was assessed through the determination of TBARS such as malondialdehyde (MDA). TBARS were determined following Ben Slima et al. [2] at days 1, 5, 10 and 15 of storage. Briefly, 2 g of sample were homogenized with 100 mL of butylated hydroxy toluene and 16 mL of trichloro acetic acid. Then, 2 mL of filtrate were added to 2 mL of thiobarbituric acid solution and heated at 70 °C for 30 min. The formation of pink color was measured at 532 nm, 508 nm and 600 nm.

#### 2.8.4. Oxymyoglobin (OxyMb) Analysis

The contents of OxyMb in sausage were determined as described by Ben Slima et al. [5]. Briefly, 5 g samples were homogenized with 25 mL ice-cold phosphate buffer and were kept for 1 h at 4 °C. The supernatant obtained was filtered and the absorbance was measured. Then, the OxyMb rates were calculated using the following formula:OxyMb (%) = [0.882 × (A_572_/A_525_) − 1.267 × (A_565_/A_525_) + 0.809 × (A_545_/A_525_) − 0.361] × 100

#### 2.8.5. Sensory Evaluation

Sensory evaluation of formulated sausages was performed by using 35 panelists from CHAHIA Company (Sfax, Tunisia). The panel was previously trained to depict the sensory features of sausages. The panelists were in a good health, nonsmokers, not color blind and had no strong likes or dislikes regarding the food to be tested. Each inspector evaluated each sub-sample with three different assays. On each analysis day, sub-samples evaluation was carried out in three sessions. The panelists scored the sensory color, texture, odor and general acceptability attributes on a 9-mark hedonic scale (9 = extremely like, 1 = extremely dislike). A score of 4 was considered as the lowest tolerable limit of acceptability.

### 2.9. Statistical Analysis

Data were expressed as mean ± standard errors (SE) and statistically analyzed using the one-way ANOVA procedure with the SPSS program (V17.0) using Tukey’s post-hoc test. If *p* < 0.05, differences were judged as statistically significant. All processes, including the manufacturing procedures were done in triplicate to validate result accuracy and process repeatability.

The formulations (control vs. reformulated), the period of storage and the interaction between them were considered as fixed effects. Meanwhile, the random effect consisted of the batch.

In order to check the validity and accuracy of results, all manipulations (formulations, experiments, analyses) were carried out in triplicate. Thus, repeatability of the processes was guaranteed. To assess the independent association of the samples’ color, textural and sensorial parameters with lipid and protein oxidation after 15 days of storage, Durbin–Watson statistic tests were used to perform multiple stepwise regression (MSR) analyses.

## 3. Results and Discussion

### 3.1. Physico-Chemical Analysis of LWSP

LWSP physical properties proved that this polysaccharide contained 76.03% of carbohydrate, 0.3% of fat and 12% of protein. It consisted of 3.83% of moisture and 7.61% of ash. Ben Slima et al. [5] demonstrated that polysaccharide–protein complexes still resided in the extract because of the hydrophobic and hydrogen interactions and covalent bonds. LWSP presented a light (L* = 66.23) and a slightly red color (a* = 0.53). This characteristic enhanced their suitability for incorporation in food.

### 3.2. Composition of LWSP

#### 3.2.1. TLC Analysis

The TLC results are illustrated in Figure 1. The hydrolysis of LWSP led to the emergence of four plugs with retention factors of 0.55, 0.60, 0.63 and 0.69 corresponding, respectively, to the standard monosaccharide glucose, mannose, arabinose and xylose. LWSP does not contain galactose, tagatose, rhamnose, and fructose in its carbohydrate composition.

#### 3.2.2. NMR Analysis

Figure 2 shows LWSP ^1^H NMR and ^13^C NMR spectra in solid state NMR at 400 and 75.5 MHz, respectively. In the ^1^H NMR spectrogram (Figure 2A), a cramped region between 3.2 to 4.41 ppm indicates several similar sugar residues which confirm the presence of polysaccharides [23]. Four proton resonance signal peaks can be observed in the anomeric proton region at 4.3, 3.9, 3.7 and 3.2 ppm, indicating four monosaccharide residues with α and β anomers. In fact, it was generally believed that the chemical shift values of α-anomeric protons were mostly larger than 4.0 ppm, while the signals less than 4.0 ppm corresponded to the β-anomeric proton [20]. In previous data, signals between 3.2 and 3.9 ppm were attributed to the characteristics of H2-H5 resonate. Therefore, the absorption signal between 3.64 and 3.94 ppm was provoked by protons on sugar rings. An intense peak signal observed at 1.0 and 1.2 ppm identifies the carbon group (R-CH2-CH3).

The ^13^C NMR spectrum showed the presence of sugar rings in our polysaccharides (Figure 2B). In fact, in this spectrogram, two anomeric carbons ofα-and β-configuration are depicted in the regions of 95.26 ppm and 110.45 ppm, respectively. This finding confirms the results of the ^1^H NMR spectrum. In addition, four anomeric carbons were observed in the regions at115 ppm, 106 ppm, 101 ppm and 95ppm in the ^13^C NMR spectrum of LWSP, which should be assigned to the anomeric carbon atoms of α-L-Ara, β-D-Xyl, β-D-Man and α-D-Glc, respectively. The high resonance signal at 20 ppm is attributed to anacetyl group. Nep and Conway [21] suggested that some sugar residues may be acetylate. The absorption signals at 59.74 and 61.17 ppm found in the ^13^C NMR spectrum can be assigned to an O–CH3 group [24]. In the ^13^C NMR spectrum of LWSP, a group of peaks collected between 75 and 83 ppm indicates that the C-2 of Man should be replaced. The peak at 75 ppm showed that Glc has a 1 → 4 bond type [25]. A strong signal found in the region 69 to 73 refers to the C-6 of Glc. Furthermore, we noticed the presence of a signal in the region ranging from 57 to 87 ppm, which can be assigned to sugars C2–C6 [26]. The signal between 69 and 77.8 corresponds to the osidic groups (C2–C5) [21]. The strong signal positioned in the region of 60–80 ppm was allocated to the pyranose configuration in LWSP.

### 3.3. Functional Properties of LWSP

The most essential functional properties in food products are related to texture enhancement. The interactions between components including water and lipid are WHC and OHC. WHC and OHC of LWSP at 25, 50 and 75 °C were investigated (Figure 3A,B). Significant discrepancies (*p* < 0.05) were observed in the values obtained at different temperatures. In this study, the highest WHC value evolved at 50 °C (14.63 ± 0.56 g H_2_O/g sample). LWSP possessed a WHC level higher than polysaccharides isolated from black cumin seeds at different temperatures [19]. In addition, the highest OHC value was obtained at 75 °C in this study, accounting for the 1.686 ± 0.16 g oil/g sample. This OHC could be related to the hydrophilic nature of the sugar monosaccharides building blocks. In addition, a high molecular weight of polysaccharides is added to thicken the displacing fluid, so as to decrease the aqueous phase mobility, enlarge the swept volume and, therefore, improve oil recovery efficiency [27]. In a previous study, LWSP was revealed as a sponge-like structure containing numerous cavities. This structure allows the LWSP to absorb a large amount of water. Such a characteristic makes it a fast swelling system in several applications like gelling and emulsifying agents [28]. In addition, a homogenous structure of the LWSP with small spherical particles was observed. Generally, the diameter of polysaccharide ranged from 10 to 40 nm.

### 3.4. Physico-Chemical Analysis of Sausages

#### 3.4.1. TPA Analyses

Texture is a predominant parameter of the acceptability and quality of meat products. Data of the main effect of different formulated sausages after 15 days of storage (4 °C) on TPA analyses are illustrated in Table 1 and Figure 4. Results showed that LWSP addition induced a significant decrease (*p* < 0.05) in hardness compared to the control. This decrease indicated softer products, thus becoming more acceptable to consumers for all samples (Ben Slima et al., 2017). Among all the texture analyses, hardness is an important parameter of freshness and quality of meat products. It varies according to several intrinsic biological factors including the connection of both collagen and fat content with muscle fiber density. The hardness values were significantly reduced with the addition of different antioxidants and reached 5.23 after 15 days of storage at 4 °C. Several studies have reported the same effects of antioxidant supplementation [29]. Polysaccharide addition might have decreased the hardness of sausage by enhancing emulsion stability through their protective role on proteins against oxidation [30]. In Roghayeh et al. [31], acceptability and softness were positively correlated. Indeed, the decrease of hardness indicated softer products, thus becoming a recurrent consumers’ determinant for purchasing. Moreover, LWSP improves the textural profile as concerns cohesiveness reduction at 15days of storage. In fact, the increase of cohesiveness can make products hard and unpleasantly sticky. Table 1 also shows that chewiness decreased with the addition of polysaccharides during storage days. In fact, the chewiness depended greatly on the concentration of polysaccharides. However, springiness did not follow any particular trend during storage in all products. Moreover, no significant differences (*p* > 0.05) were perceived between different formulations. Similar results were achieved by Lorenzo and Franco [32] who found that this parameter resisted antioxidant supplementation in sausages. In addition, Tee and Siow [33] reported that this parameter was also unchanged with the presence of starch. 

#### 3.4.2. Color Measurements

Data of the main effect of different formulated sausages after 15 days of storage (4 °C) on instrumental color (L*, a*, b*) are illustrated in Table 2. The samples’ lightness increased in parallel with the storage time and their darkness with the antioxidant addition. Indeed, the increasing light scattering during storage could be linked to protein coagulation, which liberates water to the surface [34]. L* was meanwhile altered by the presence of antioxidant ingredients. The negative control (T1) had a relatively higher L* value than the other products. The results corroborate the work of Plagarini et al. [35]. However, the control sausage formulated with ascorbic acid had a higher a* value in comparison with the negative control from the 1stuntil the last day of storage. The increase of a* value reflects the favorable role in preserving the redness of sausages instead of it being decreased through oxidation and deterioration. Other authors reported that samples with antioxidants increased the a* value [1,35]. The red color is attributed to the presence of oxymyoglobin, an oxygenated myoglobin. The retardation of metmyoglobin formation can be assigned to the antioxidant compounds in the polysaccharide [1], which detects the ability of transferring one electron to reduce compounds that can also be carbonyls, metals and radicals. It results in a change in color when this compound is reduced. The molecular weight along with other chemicals present in polysaccharide fractions such as the carboxyl group can also be responsible for antioxidant activities [36]. The yellowness of the formulated sausages decreased with antioxidant treatments and significantly increased with the storage period. In addition, a b* value decline of meat product containing natural antioxidants was indicated by Rojas and Brewer [37]. According to Jiang et al. [38], lower b* values refer to less pale meat, which is preferred by consumers. An analysis of ΔE revealed the most pronounced changes in color between samples.

### 3.5. TBARS

The assessment of TBARS serves to determine the lipid oxidation degree. Data of the main effect of different formulated sausages after 15 days of storage (4 °C) on TBARS values are illustrated in Figure 5A. TBARS values increased from 0.2 to 1 mg MDA/kg during storage in all samples. TBARS values are linked with sensorial perception of lipid oxidation. Indeed, values superior to 1 mg MDA/kg indicated “off-odors” formation [32]. On the 15th day of storage, TBARS content decreased in the reformulated sausages. Samples without antioxidant addition showed more intense lipid oxidation. These results prove that crude polysaccharide reduced TBARS values, which is in agreement with results conveyed by several authors [5,6]. Polysaccharides extracted from different sources have antioxidant properties and consequently reduced lipid deterioration (TBARS) through the inhibition of MDA formation during storage. Interestingly, LWSP addition can prolong the shelf-life of sausages as compared to ascorbic acid, favoring its application in food industries. In fact, certain polysaccharides described in the literature [39,40] are able to reduce lipid peroxidation. These results suggest that LWSP, excellent proton and electron donors, act on free radicals to convert them to more stable products.

### 3.6. Effect on OxyMb Oxidation

Myoglobin chemical state reflects meat color. Indeed, the red coloration in meat is attributed to the presence of OxyMb, which is an oxygenated myoglobin. Data of the main effect of different formulated sausages after 15 days of storage (4 °C) on OxyMb content are illustrated in Figure 5B. On the first day, OxyMb content in all samples of sausages ranged between 51% and 52%. After 4 days of refrigerated storage, OxyMb content decreased (*p* < 0.05) in all samples, reaching 10.5% in the control. OxyMb oxidation decreased in sausages formulated with antioxidants (ascorbic acid or LWSP) compared to control. Thus, antioxidants were used to slow down oxidation processes. The OxyMb decreased gradually by losing an electron, thus contributing to the brown coloration of the meat in sausages [41]. Falowo et al. [42] reported that synthetic or natural compounds are regarded as compounds that promote OxyMb and deoxymyoglobin pigments formation in fresh meat. Similarly, Ben Slima et al. [5] demonstrated that polysaccharides exhibiting antioxidant activities were able to protect myoglobin against oxidation processes and to maintain myoglobin stability. This stability was probably the result of the interruption of free radical chain propagation or the inhibition of free radicals formation. Thus, LWSP can enhance shelf-life characteristics of sausages.

### 3.7. Sensory Analysis

Sensory evaluation is an interesting method for judging the quality of sausages in terms of color, texture, odor and general acceptability. Data of the main effect of different formulated sausages after 15 days of storage (4 °C) on sensory analysis are illustrated in Table 3.

On the first day of storage, sausages prepared using LWSP did not present significant differences compared to the control in all analysis parameters. After 15 days of refrigerated storage, the texture, color, and general acceptability (GA) were lowest in sausages without antioxidant addition compared to treated sausages. In fact, the sausage samples prepared using only LWSP had the highest scores in all parameters. Thus, the panelists mostly disliked the control product due to oxidative damage. Indeed, changes in meat color are due to the oxidation of red oxymyoglobin to metmyoglobin, which gives rise to an unattractive brown color [1]. In addition, polysaccharides have been shown to affect the texture of foods. Regarding the overall acceptability of the studied samples, it was found that on the first day of storage, the control sample scored (9.00); then, this overall acceptability decreased gradually. Similar results were obtained by Kallel et al. [1], who reported that beef meat treated with garlic straw polysaccharide was more acceptable than control.

Association of protein and lipid oxidation with texture profile, color parameters, and sensorial evaluation changes in samples of sausages

The independent association between protein and lipid oxidation with different parameters after a period of 15 days of storage was evaluated through MSR analyses in a model including hardness, chewiness, cohesiveness, springiness, a*, b*, L*, odor, overall color, texture and GA as independent variables (Table 3).

OxyMb, cohesiveness, hardness, texture, GA and a* independently varied in association with TBARS (*p* < 0.0001). As for TBARS, a significant correlation was observed with cohesiveness and hardness. These results were similar to Smaoui et al. [43], who asserted that TBARS was linked to instrumental texture, especially hardness, cohesiveness and springiness. Similarly, TBARS was in association with OxyMb. Indeed, Chan et al. [44] demonstrated that secondary lipid oxidation products, mainly aldehydes, were pro-oxidative towards the oxidation of oxymyoglobin. It has been suggested that peroxides derived from lipid oxidation or free radicals could react with myoglobin and cause it to oxidize. Other researchers such as Zakrys et al. [45] found a negative correlation of TBARS and oxymyoglobin. TBARS values increased with the decrease of OxyMb. In addition, the presence of unsaturated fatty acids in meat is related to lipid oxidation [46]. 

Meanwhile, OxyMb was evaluated through TBARS, L*, a*, b*, chewiness, color and GA. Oxymyoglobin, the main pigment in meat tissues, is responsible for its bright red color. Concerning the association of OxyMb with L*, a*, and b*, myoglobin presents three forms: deoxymyoglobin (Mb) or reduced myoglobin, oxymyoglobin (OxyMb) or oxygenated myoglobin and metmyoglobin (MetMb) or oxidized myoglobin, which have a purple red color, a bright red color and a brown color, respectively. Consumers find the bright red color associated with OxyMb formation desirable in meat products [47]. This reaction could explain the correlation between OxyMb and GA. Protein oxidation effects on textural parameters have been further studied in meat. Indeed, less protein oxidation leads to fewer cross-links and subsequently less texture deterioration. This combination explains the relation between OxyMb and chewiness [48].

## 4. Conclusions

This study was undertaken to characterize a novel water-soluble polysaccharide extracted from flaxseed (LWSP). It aims equally to evaluate its functional properties and antioxidant activities. Interestingly, LWSP addition successfully enhanced the oxidation stability of proteins and lipids during 15 days of storage at 4 °C. The correlations between TBARS, OxyMb, texture profile (hardness, cohesiveness, chewiness and springiness), color parameters (a*, b*, and L*) and sensorial properties of different formulated products were examined. Remarkably, LWSP incorporation in sausages improved color and textural parameters (hardness, cohesiveness and chewiness) due to the formation of a LWSP-meat protein matrix. Sensory analysis showed that LWSP led to the development of better sensory quality with a markedly superior global acceptability. This polysaccharide could be an efficient ingredient for future applications in several food industries.

## Figures and Tables

**Figure 1 polymers-15-01014-f001:**
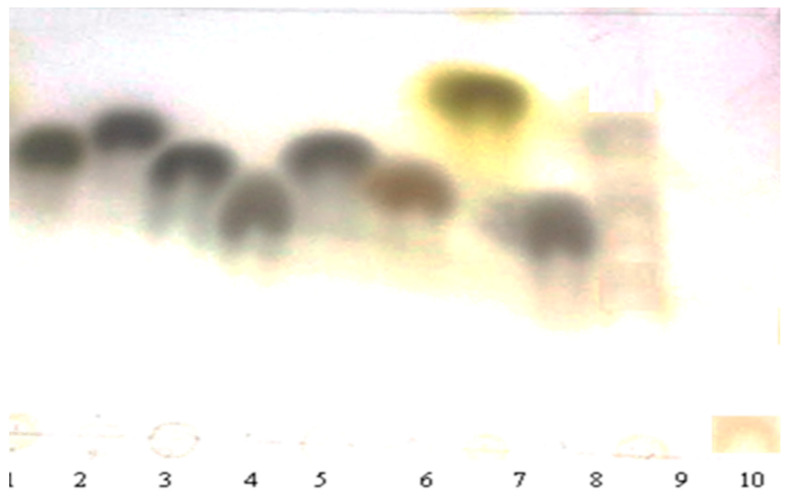
TLC analysis of LWSP. 1. Arabinose, 2. Xylose, 3. Fructose, 4. Glucose, 5. Tagatose, 6. Mannose, 7. Rhamnose, 8. Galactose, 9. LWSP hydrolyzed via TFA, 10. Unhydrolyzed LWSP.

**Figure 2 polymers-15-01014-f002:**
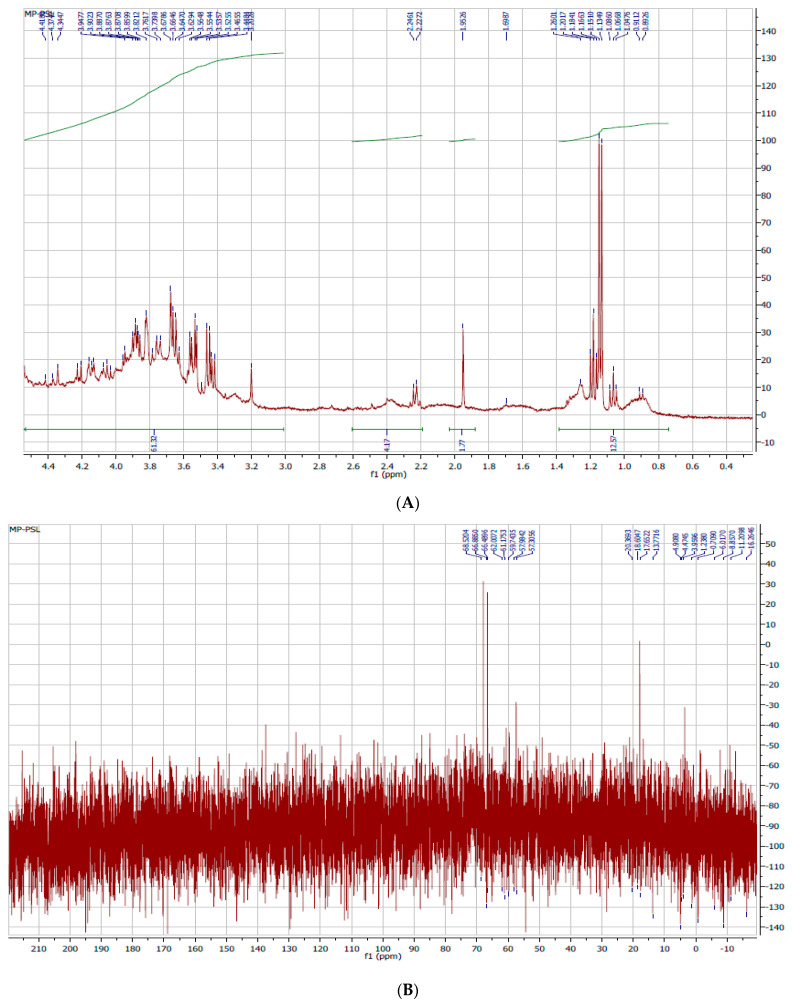
(**A**) ^1^H NMR spectrum of LWSP in D_2_O at 400 MHz and 25 °C, (**B**): solid state ^13^CP/MAS NMR spectrum of LWSP at 75.5 MHz and 25 °C.

**Figure 3 polymers-15-01014-f003:**
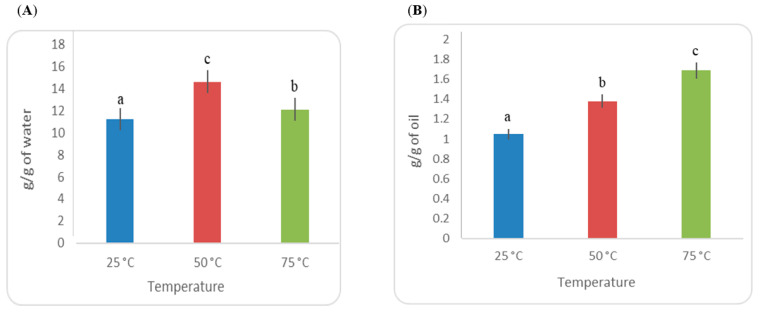
Evolution of the WHC (**A**) and OHC (**B**) of LWSP as a function of temperature (25, 50, and 75 °C). Different letters refer to significant differences (*p* < 0.05).

**Figure 4 polymers-15-01014-f004:**
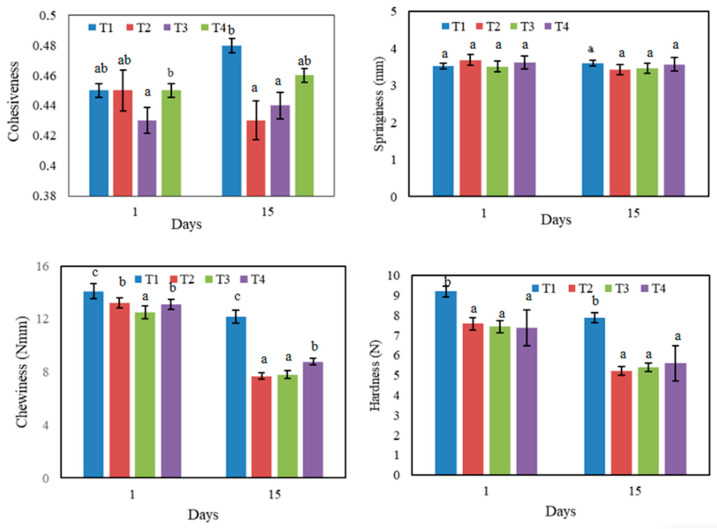
TPA parameters of different sausages during refrigerated storage. a, b, c: Different letters are significantly different *p* < 0.05 between formulations.

**Figure 5 polymers-15-01014-f005:**
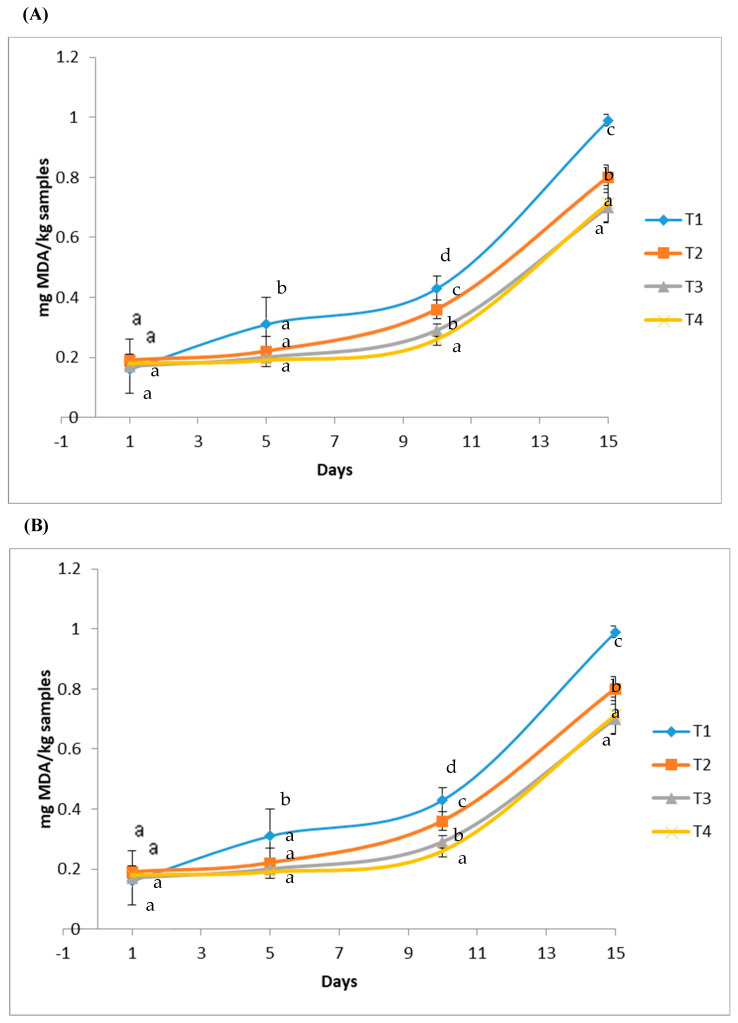
(**A**) Lipid oxidation (TBARS mg MDA/kg sample) values of different sausages during refrigerated storage. (**B**) Oxymyoglobin (%) of different sausages during refrigerated storage. T1: control sausage which did not contain either LWSP or ascorbic acid; T2: standard formulation consisting of 0.125% of ascorbic acid; T3: sausage reformulated with LWSP at 0.125% and T4: sausage reformulated with LWSP at 0.062% and ascorbic acid at 0.062%. Different letters refer to significant differences (*p* < 0.05) between samples.

**Table 1 polymers-15-01014-t001:** Texture profile analysis of sausages during refrigerated storage.

Texture Parameters	Days	T1 (Control)	T2 (0.125% of Ascorbic Acid)	T3 (0.125% of LWSP)	T4 (0.062% of Ascorbic Acid and 0.062% of LWSP)
Cohesiveness	1	0.45 ± 0.02 ^ab1^	0.45 ± 0.01 ^ab2^	0.43 ± 0.01 ^a1^	0.45 ± 0.01 ^b1^
	15	0.48 ± 0.02 ^b2^	0.43 ± 0.03 ^a1,2^	0.44 ± 0.02 ^a1^	0.46 ± 0.01 ^ab1^
Springiness (mm)	1	3.52 ± 0.16 ^a1^	3.68 ± 0.25 ^a1^	3.51 ± 0.58 ^a1,2^	3.62 ± 0.20 ^a2^
	15	3.60 ± 0.06 ^a1,2^	3.42 ± 0.22 ^a2^	3.47 ± 0.13 ^a1^	3.57 ± 0.05 ^a1,2^
Hardness (N)	1	9.20 ± 0.30 ^b2,1^	7.59 ± 0.36 ^a2,1^	7.44 ± 0.22 ^a2,1^	7.39 ± 0.02 ^a2,1^
	15	7.88 ± 0.44 ^b2,1^	5.23 ± 0.37 ^a2,1^	5.39 ± 0.13 ^a2,1^	5.60 ± 0.43 ^a2,1^
Chewiness (Nmm)	1	14.10 ± 0.11 ^c2,1^	13.20 ± 0.12 ^b2,1^	12.50 ± 0.13 ^a2,1^	13.10 ± 0.13 ^b2,1^
	15	12.20 ± 0.05 ^c2,1^	7.69 ± 0.22 ^a2,1^	7.80 ± 0.07 ^a2,1^	8.78 ± 0.16 ^b2,1^

T1: control sausage which did not contain either LWSP or ascorbic acid; T2: standard formulation consisting of 0.125% of ascorbic acid; T3: sausage reformulated with LWSP at 0.125% and T4: sausage reformulated with LWSP at 0.062% and ascorbic acid at 0.062%. Values are given as mean ± SE from triplicate determinations (*n* = 3). Different letters (a, b, c) mean significant differences between samples within a row (*p* < 0.05) and different numbers (1, 2) in the same column indicate significant differences during storage period (*p* < 0.05) for the same sample. Beef sausages were formulated in triplicate.

**Table 2 polymers-15-01014-t002:** Color parameters (L*, a*, b*) of sausages during refrigerated storage.

Color Parameters	Days	T1 (Control)	T2 (0.125% of Ascorbic Acid)	T3 (0.125% of LWSP)	T4 (0.062% of Ascorbic Acid and 0.062% of LWSP)
L*	1	41.71 ± 1.72 ^a1^	43.23 ± 0.13 ^ab1^	43.46 ± 1.20 ^ab1,2^	44.23 ± 0.38 ^b1,2^
	15	43.00 ± 0.99 ^a2^	46.94 ± 0.79 ^c2^	44.77 ± 0.49 ^b1,2^	45.36 ± 0.60 ^bc1,2^
a*	1	15.14 ± 0.13 ^a2^	16.47 ± 0.50 ^b1^	16.08 ± 0.87 ^ab1^	16.07 ± 0.49 ^ab1^
	15	13.08 ± 0.44 ^a1^	17.06 ± 0.11 ^c1,2^	17.33 ± 0.28 ^c1,2^	16.07 ± 0.49 ^b1^
b*	1	13.96 ± 0.03 ^b1^	13.19 ± 1.08 ^ab1^	12.58 ± 0.30 ^a1^	12.54 ± 0.21 ^a1^
	15	15.64 ± 0.61 ^b2^	13.55 ± 0.53 ^a1^	13.69 ± 0.32 ^a1,2^	13.18 ± 0.11 ^a2^
ΔE	1	-	2.16 ± 0.01 ^b1^	0.76 ± 0.04 ^a1^	0.77 ± 0.05 ^a1^
	15	-	5.98 ± 0.03 ^c2^	2.19 ± 0.02 ^b2^	1.48 ± 0.08 ^a2^

T1: control sausage which did not contain either LWSP or ascorbic acid; T2: standard formulation consisting of 0.125% of ascorbic acid; T3: sausage reformulated with LWSP at 0.125% and T4: sausage reformulated with LWSP at 0.062% and ascorbic acid at 0.062%. Values are given as mean ± SE from triplicate determinations (*n* = 3). Different letters (a, b, c) mean significant differences between samples (*p* < 0.05) within a row and different numbers (1, 2) in the same column indicate significant differences during storage period (*p* < 0.05) for the same sample. Beef sausages were formulated in triplicate.

**Table 3 polymers-15-01014-t003:** A. Sensory analysis of sausage during refrigerated storage. B. Stepwise multiple linear regression analysis between lipid and protein oxidation and color and texture parameters at 15 days of storage.

Sensory Analysis	Days	T1 (Control)	T2 (0.125% of Ascorbic Acid)	T3 (0.125% of LWSP)	T4 (0.062% of Ascorbic Acid and 0.062% of LWSP)
Odor	1	7.86 ± 0.52 ^a2^	8.02 ± 0.36 ^a3^	7.94 ± 0.59 ^a2^	8.00 ± 0.99 ^a2^
	5	7.42 ± 0.71 ^a2^	7.63 ± 0.70 ^a2,3^	7.28 ± 0.71 ^a2^	7.58 ± 1.09 ^a1,2^
	10	7.00 ± 0.79 ^a1,2^	6.80 ± 0.57 ^a2^	7.00 ± 0.51 ^a2^	7.09 ± 0.7 ^a1,2^
	15	5.80 ± 0.63 ^a1^	5.90 ± 0.07 ^a1^	6.00 ± 0.29 ^a1^	6.14 ± 0.45 ^a1^
Color	1	6.50 ± 0.33 ^a2^	8.87 ± 0.29 ^b2^	9.27 ± 0.51 ^b3^	7.26 ± 0.59 ^a1^
	5	6.60 ± 0.71 ^a2^	8.72 ± 0.54 ^bc2^	8.88 ± 0.35 ^c2,3^	7.90 ± 0.21 ^b1^
	10	6.20 ± 0.63 ^a1,2^	8.68 ± 0.47 ^b1^	8.40 ± 0.33 ^b2^	7.80 ± 0.63 ^b1^
	15	5.36 ± 0.69 ^a1^	8.14 ± 0.39 ^b2^	8.25 ± 0.72 ^b1^	7.45 ± 1.32 ^b1^
Texture	1	7.38 ± 0.54 ^a1,2^	8.00 ± 0.75 ^a2^	8.24 ± 0.59 ^a1^	8.31 ± 0.70 ^a1^
	5	7.40 ± 0.41 ^a2^	7.77 ± 0.49 ^ab2^	8.00 ± 0.32 ^ab1^	8.51 ± 0.40 ^b1^
	10	6.90 ± 0.86 ^a1,2^	7.04 ± 0.39 ^ab1,2^	7.99 ± 0.32 ^bc1^	8.50 ± 0.39 ^c1^
	15	6.20 ± 0.65 ^a1^	6.60 ± 0.63 ^a1^	8.10 ± 0.63 ^b1^	8.40 ± 0.28 ^b1^
General acceptability	1	7.50 ± 0.51 ^a2^	9.00 ± 0.62 ^b2^	9.30 ± 0.23 ^b2,3^	9.02 ± 0.17 ^b1^
	5	7.00 ± 0.64 ^a2^	8.42 ± 0.35 ^b2^	9.00 ± 0.71 ^b2^	8.70 ± 0.63 ^b1^
	10	6.20 ± 0.39 ^a1,2^	7.60 ± 0.60 ^b1^	8.00 ± 0.14 ^b1^	7.80 ± 0.39 ^b1^
	15	6.02 ± 0.63 ^a1^	7.00 ± 0.70 ^a2^	8.50 ± 0.51 ^b^	8.20 ± 0.63 ^b1^
Dependent Variables			Independent Variables	Β	*p* Value
TBARS			samples	−0.124	<0.0001
R^2^ = 0.999			OxyMb	0.047	<0.0001
R^2^ adjusted = 0.997			Cohesiveness	0.355	<0.0001
			a*	−0.049	<0.0001
			b*	−0.066	0.083
			L*	0.026	0.121
			Springiness	0.005	0.693
			Chewiness	0.1	0.204
			Hardness	0.084	<0.001
			Odor	0.022	0.146
			Texture	−0.116	<0.0001
			Color	−0.075	0.063
			GA	−0.113	<0.0001
OxyMb			Samples	2.670	<0.0001
R^2^ = 0.821			a*	0.495	<0.0001
R^2^ adjusted = 0.812			L*	0.369	<0.0001
			b*	−0.414	0.005
			Cohesiveness	−0.247	0.010
			Springiness	−0.157	0.117
			Hardness	−0.403	0.006
			Chewiness	−0.494	<0.001
			Odor	0.091	0.404
			Texture	−0.013	0.947
			Color	0.470	<0.001
			GA	0.328	0.041
			TBARS	−0.403	0.036

T1: control sausage which did not contain either LWSP or ascorbic acid; T2: standard formulation consisting of 0.125% of ascorbic acid; T3: sausage reformulated with LWSP at 0.125% and T4: sausage reformulated with LWSP at 0.062% and ascorbic acid at 0.062%. Values are given as mean ± SE from triplicate determinations (*n* = 3). Different letters (a, b, c) mean significant differences between samples (*p* < 0.05) within a row and different numbers (1, 2) in the same column indicate significant differences during storage period (*p* < 0.05) for the same sample. Beef sausages were formulated in triplicate.

## Data Availability

The data presented in this study are available on request from the corresponding authors.

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
