# Peer review of "Effects of a Novel Lin Seed Polysaccharide on Beef Sausage Properties"

_polymers, 2023, doi:10.3390/polym15041014_

Round 1
Reviewer 1 Report
The manuscript Effects of a novel Lin seeds polysaccharide on beef sausages properties is well written and organized. To obtain better quality of paper could you in Introduction explain more about physicochemical properties of LWSP, provide a full name and structure.Author Response
-The manuscript Effects of a novel Lin seeds polysaccharide on beef sausages properties is well written and organized. To obtain better quality of paper could you in Introduction explain more about physicochemical properties of LWSP, provide a full name and structure.
-As recommended, we have added this data in Introduction section (Please see Introduction section).

Reviewer 2 Report
The purified water-soluble polysaccharide (LWSP) played an important role in sausage quality. The author carried out texture analysis, color analysis, sensory analysis and oxidation property evaluation of the purified polysaccharide, which resulted in a heavy workload.
But there are problems with the manuscript as follows:
Page 4 Format for thiobarbiturate analysis.
Figure 1 is not clear.
In Figure 3 (B), 75°C obtained the highest OHC value in the figure, but there was no turning point in the chart, so the explanation was somewhat far-fetched.
As mentioned on page 8, the hardness values were significantly reduced after the addition of different antioxidants, but the hardness values of the control group were also significantly decreased, which cannot prove that the effect was caused by the addition (LWSP).
Table 1 has a formatting problem.
Page 9, (LWSP) improved the structural profile and reduced Cohesiveness . This is not evident in Table 1.
On the eleventh page, the content of TBARS says that T3>T4, but it cannot be reflected in Figure 5 (A).
Page 13. At the end of storage, LWSP is more effective than ascorbic acid. This is not evident in Figure 5 (B).

Author Response
-Page 4 Format for thiobarbiturate analysis.
As recommended, we have changed the format for thiobarbiturate analysis (Please see page 4).
-Figure 1 is not clear.
We have improved the quality of Figure (Please see Figure 1).
-In Figure 3 (B), 75°C obtained the highest OHC value in the figure, but there was no turning point in the chart, so the explanation was somewhat far-fetched.
We have more explained this part (please see page 8).
-As mentioned on page 8, the hardness values were significantly reduced after the addition of different antioxidants, but the hardness values of the control group were also significantly decreased, which cannot prove that the effect was caused by the addition (LWSP).
The hardness values were reduced in reformulated sausages compared to the control. However the reduction of hardness during storage period were observed in control and reformulated
sausages
-Table 1 has a formatting problem.
We have corrected the format of Table 1 (Please see Table 1)
-Page 9, (LWSP) improved the structural profile and reduced Cohesiveness. This is not evident in Table 1.
We thank the reviewer at this comment. In fact, we have observed the reduction of cohesiveness just at the end of storage. We have added this remark in the text (Please see page 9).
-On the eleventh page, the content of TBARS says that T3>T4, but it cannot be reflected in Figure 5 (A).
We have corrected this phrase (Please see page 11).
-Page 13. At the end of storage, LWSP is more effective than ascorbic acid. This is not evident in Figure 5 (B).
We have eliminated this phrase (Please see page 13).

Reviewer 3 Report
Polymers
polymers-2077644
Effects of a novel Lin seeds polysaccharide on beef sausages properties
Dear Editor,
The article deals with the determination of the effect of the incorporation of novel Lin seeds polysaccharide in beef sausages on lipid and myoglobin oxidation, as well as physicochemical, color, texture and sensory characteristics. The paper has been well designed and written. My specific comments and questions are below;
- Please give more information about the Lin seeds polysaccharide in the introduction section,
- Material and methods: please use g value instead of rpm for centrifuge process
- Please give more information about the TBARS and oxymyoglobin analyses!
- The overall LWSP physicochemical analyses result is higher than 100%. Can you check the results?
Author Response
The article deals with the determination of the effect of the incorporation of novel Lin seeds polysaccharide in beef sausages on lipid and myoglobin oxidation, as well as physicochemical, color, texture and sensory characteristics. The paper has been well designed and written. My specific comments and questions are below;
- Please give more information about the Lin seeds polysaccharide in the introduction section,
As suggested, we have added more information about Lin seeds polysaccharides (Please see Introduction section).
- Material and methods: please use g value instead of rpm for centrifuge process.
We have used g value for centrifuge process (Please see Materials and Methods).
- Please give more information about the TBARS and oxymyoglobin analyses!
We have more explained the TBARS and OxyMb analysis (Please see page 5).
- The overall LWSP physicochemical analyses result is higher than 100%. Can you check the results?
We thank the reviewer of this pertinent remark. Indeed, we have verified the percentage (Please see page 5).
